# Waymarking in Social Robots: Environment Signaling Using Human–Robot Interaction

**DOI:** 10.3390/s21238145

**Published:** 2021-12-06

**Authors:** Ana Corrales-Paredes, María Malfaz, Verónica Egido-García, Miguel A. Salichs

**Affiliations:** 1Science, Computation and Technology Department, School of Architecture, Engineering and Design, Universidad Europea de Madrid, 28670 Villaviciosa de Odón, Spain; 2Robotics Lab, Universidad Carlos III de Madrid, Av. de la Universidad 30, Leganés, 28911 Madrid, Spain; mmalfaz@ing.uc3m.es (M.M.); salichs@ing.uc3m.es (M.A.S.); 3Vice-Dean Engineering, School of Architecture, Engineering and Design, Universidad Europea de Madrid, 28670 Villaviciosa de Odón, Spain; veronica.egido@universidadeuropea.es

**Keywords:** human–robot interaction, social robots, waymarking, radio frequency identification, RFID, signaling, navigation, wayfinding, topological, indoor

## Abstract

Travellers use the term waymarking to define the action of posting signs, or waymarks, along a route. These marks are intended to be points of reference during navigation for the environment. In this research, we will define waymarking as the skill of a robot to signal the environment or generate information to facilitate localization and navigation, both for its own use and for other robots as well. We present an automated environment signaling system using human–robot interaction and radio frequency identification (RFID) technology. The goal is for the robot, through human–robot interaction, to obtain information from the environment and use this information to carry out the signaling or waymarking process. HRI will play a key role in the signaling process since this type of communication makes it possible to exchange more specific and enriching information. The robot uses common phrases such as “Where am I?” and “Where can I go?”, just as we humans do when we ask other people for information about the environment. It is also possible to guide the robot and “show” it the environment to carry out the task of writing the signs. The robot will use the information received to create, update, or improve the navigation data in the RFID signals. In this paper, the signaling process will be described, how the robot acquires the information for signals, writing and updating process and finally, the implementation and integration in a real social robot in a real indoor environment.

## 1. Introduction

The autonomy of robots is one of the main researchers objectives in recent decades [1,2]. One of the basic tasks to achieve autonomy for a robot is to move in unknown environments and reach a given destination, just as we humans do. In humans, this process is called “Wayfinding” [3]. In robotics, the term known is “Navigation” [4,5,6,7]. Studying the navigation of social robots is a complex issue involving the robot’s capabilities, the human, the proxemics, the social context, and the environment [8].

It is common for people to ask for information when we want to get to a place and we have no other elements to guide us towards the goal. In robotics, there are some works that use human–robot dialogues to give route instructions to a robot so that it can reach a destination [9,10,11,12,13]. Some researches use semantic and cognitive maps [14,15,16,17] and dialogues with basic directions [18,19] or include a SLAM algorithm for navigation and geometric localization [20]. However, in these semantic or topological maps, human–robot interaction does not play a key role in their building process.

In areas such as ecology, psychology and geography, the term “Waymarking” is used to define the action of posting signs, or waymarks, along a route so that travellers can easily follow the route [21]. These marks serve as points of reference during exploration and navigation for the environment. Several research projects include indirectly this concept to refer to the information generated by a robot for its own use or for communication with other robots [22,23,24,25].

The fact that a robot can signal the environment makes it possible that the information on the signals is always updated. This capability can be helpful for waymark from scratch or if there are changes in the environment. Let us give an example: Imagine a robot in a building. The robot will interact with users to obtain information from the environment and carry out the waymarking process. Questions such as: Where am I? Where can I go? or following a set of instructions to get to a place will help the robot expand its knowledge. The robot could create and update autonomously the signaling system data for its navigation or others robots.

Thanks to human–robot interaction, a person may communicate naturally with the robot and indicate where it is and where it can go. The robot can use this information to update the signals data.

This article presents an automated environment signaling system using human–robot interaction and radio frequency identification (RFID) technology. The goal is for the robot, through human–robot interaction, to obtain information from the environment and use this information to carry out the signaling or waymarking process. The robot will use the information received to create, update or improve the information in the RFID signals, either for its use or for other robots that, in the future, navigate by the environment.

The system was implemented in a real indoor environment and successfully tested on the social robot Maggie [26].

The signaling system used, its design and the structure of each signal were defined in a previous paper [27]. In that research, the robot used the RFID tags to guide itself and, in case of getting lost or not reading a signal, the robot kept navigating using other signals to orientate itself again. The system facilitate the navigation of the robot in indoor environments, such as homes and buildings, in situations where it has no previous knowledge about those places.

The specific objectives of this research are the following:Design a system for writing signals in indoor environments. When the robot finds a signal incomplete or empty, it will add and/or update the information contained in the signal.To carry out the signaling process, the robot must acquire the information that it will write on the signal. To this aim, human–robot interaction must be added to the system, specifically verbal communication. This type of communication makes it possible to exchange more specific and enriching information.To implement the waymarking system in a real social robot.

This paper is organized as follows: In Section 2, we present a synopsis of the related work in robotics navigation using RFID technology, navigation with human–robot interaction and signaling or waymarking in robotics. Section 3 introduces the system overview, the social robot Maggie and its RFID signals system. Next, in Section 4, the reading and writing RFID skills are described. In Section 5, Human–Robot Interaction cases are specified. Then, in Section 6, grammars and dialogues design are shown. In Section 7, the signaling and waymarking system is described. Later, Section 8 shows the experimental results with the robot Maggie in a real environment. Finally, in Section 9, the main conclusions and some future works are presented.

## 2. Related Work

Radio frequency identification (RFID) is the wireless non-contact use of electromagnetic or electrostatic waves to transfer data [28]. An RFID system comprises three basic elements: a reader, tags and antennas.

A reader or transceiver transmits and receives waves that it transforms into digital information. Read and write operations follow the master-slave principle; the reader is activated just when receiving a software application request. Similarly, tags only respond when the reader initiates communication with them [29].A tag or RFID trasponder stores and sends information to an RFID reader. A tag includes an integrated circuit with a memory and an antenna.An antenna converts the RFID reader’s signal into RF waves, activating the tags within its reading range. There are two main types of RFID tags: An active RFID tag has its own power source, often a battery. A passive RFID tag receives its power from the reading antenna.

### 2.1. Advantages and Limitations of RFID Systems

The RFID systems have a number of advantages and limitations, as highlighted below:

#### 2.1.1. Advantages

Tags can resist harsh environmental conditions, e.g., humidity, dirt and drastic temperature changes.Fast response time (<100 ms), a reader can read hundreds of tags in short time.RFID tags can be read without the need for physical contact with a reader.Tags can be written a large number of times.The read range is variable, from a few centimetres (high frequency) to more than 100 m (ultra high frequency).Memory capacity. Tags with large memory capacities enable more detailed data.

#### 2.1.2. Limitations

Reading efficiency is proportional to the size of the antenna and the size of the tags. Small tags need large antennas and vice versa.The radio waves are attenuated when they pass through certain materials (particularly water), which are reflected when they collide with metals. This significantly reduces the reading radius.The lifetime of active tags depends on the life of their batteries.

### 2.2. RFID and Robotics Navigation

In robotics, RFID systems are a useful tool to enrich a robot’s information from the environment. These systems are efficient in the process of identifying objects or people and give support to other sensory systems of a robot, e.g., identification using vision [30,31].

The use of RFID technology in mobile robotics is focused on localization and navigation [32,33,34,35]. A robot can read information from tags either to know its orientation concerning other elements [36,37] to know where it is [38,39], or with a set of tags to mark the path to follow.

In [40,41], the authors analyse different methods of indoor localization using passive high frequency—HF and ultra high frequency—UHF RFID technology; these works describe three main categories of vehicle localization systems: (i) solutions only using RFID technology, (ii) sensor-fusion techniques combining data from RFID systems and proprioceptive sensors (odometry data), and (iii) sensor-fusion techniques combing RFID data with those of other exteroceptive sensors (e.g., Laser Range finders, computer vision), concluding that implementing a sensor-fusion system based on RFID UHF is a relevant solution applied to the indoor localization problem. [27,42,43].

In recent papers, we also find applications using SLAM algorithms fused with UHF-RFID data [44,45,46,47,48,49], RFID + SLAM algorithms allow optimizing the localization problem—both in time and accuracy—in indoor environments.

### 2.3. Navigation and Human–Robot Interaction

Other research focuses on RFID + robotics applications to support people with functional diversity or elderly people. These works use human–robot communication to enable a robot to successfully reach the set goal for navigation or help the user achieve their destination. There are several RFID applications with robots and people with visual impairment, such as shopping assistance [50,51] and obstacles detection. Kulyukin [52] presents a system to help the elderly using an intelligent wheeled walker called iWalker. RFID tags are placed under carpets or mats. The interface gives the user auditory indications such as “turn right” or “now move forward”. Routes through the environment are pre-coded and require a map of the environment. Kulyukin also proposed a shopping assistance system [50], with the purpose to assist visually impaired customers in navigating the store and carrying the purchased items around the store and to the check-out registers. The user interface input used was a hand-held keypad. The user interface output mode used non-verbal audio beacons and speech synthesis. A path planner with a previous map also was required in this approximation. The environment was represented as a graph where nodes represented the RFID tags and the edges described the behaviors needed to move between tags.

Multimodal interaction is also into the assistive robotic shopping cart for people with hearing impairments, presented in [53]. This work focuses on Russian sign language training through a single-handed gesture recognition and a touch control screen. The system uses a FastSLAM algorithm and, similar to the works seen previously, requires a prior mapping of the environment.

The combination of gestures and dialogues is also present in the ACE (autonomous city explorer) project [9], a robot navigates through outdoor environments and, by asking pedestrians questions, successfully reaches its goal. This robot uses voice-based interaction, gestures and touch-screen communication. To obtain information, it uses colloquial expressions and combines this with gestural details by asking the user to point in the direction to go. The robot navigates to a designated goal location without any prior map knowledge or GPS sensors, only by asking passers-by for the way. The main drawback of this approach was that, although the robot managed to reach the goal, it needed many interactions with different users, leading to each path taking longer than expected. One of the causes of this was that the gesture and speech recognition system was not robust enough, and the natural language interaction was not achieved.

The naturality of the interaction and the quality of information given to the robot have been studied in [11]. The authors have designed a dialogue system for giving route instructions to a robot. They base their study on a comparison between human-human interaction and human–robot interaction. The authors aim to analyse the level and quality of information (granularity) that a user expects from a machine to guide him to his destination in the navigation task and the knowledge that a user can provide to a machine. In their research, they conclude that while giving guidance instructions to a human is about achieving a higher level of information detail, to a machine the user prefers to give basic movement instructions such as “turn right” or “continue forward” so that the instructions given to a machine specify the step-by-step movements during navigation.

Other studies have focused on developing a language to give instructions to the robot [54,55]. These researches present a spatial language, called route instruction language (RIL), to describe the route between the start and the end points of a humanoid robot’s navigation task. RIL is intended as a semi-formal language for instructing robots, to be used by non-expert users via a structured GUI. For navigation tasks, the authors generate a topological map from this language of instructions that they have called formal route instructions (FRIs). Each FRI provides elementary instructions in a formal language (RIL) that are interpreted as navigation instructions internally in the robot system. In FRIs, there are three types of commands: (i) position commands, (ii) locomotion commands. (iii) orientation change commands. Although such languages give more accurate information to the robot, the natural interaction with the user is lost.

### 2.4. Waymarking and Robotics

According to the *Macmillan Dictionary*, a waymark is defined as “a mark put on a tree, wall, or other surface to show the direction that a path takes”. Waymarking is a term used to define the action of posting signs, or waymarks, on a path to guide other travellers.

The Waymarking concept can also be applied in robotics. Several research projects indirectly include this concept for localization and navigation in mobile robotics. Though, the term is not explicitly defined in robotics.

In this research we will define Waymarking as the skill of a robot to signal the environment or generate information to facilitate localization and navigation, both for its own use and that of other robots.

Some researches are inspired by the behavior of humans or animals in nature. For example, the use of pheromones to signal the environment to warn other animals, either to attract them to a place or to repel them [22]. In [23] probabilistic algorithms based on localization accuracy are used to generate these pheromones. By generating a map, information on traversable areas and sub-areas will be stored to work without interference from other objects or robots. Other research has used messaging scheme throughout the robot swarms to communicate navigational information [24] by example, to support others robots in obstacles avoidance tasks [25].

Other works use RFID tags to store data about artificial pheronomone models [56]; data stored is the algorithm input to realize the pheromone field by using an artificial pheromone system.

The RFID technology also is useful to establish communication between handling robots [57]; workpieces are equipped with read/write RFID transponders. Data stored is used to improve the robot task with respect to cycle time and energy consumption. According to the study, the incorporation of robot-to-robot communication using RFID technology, was effective for optimizing tasks such as: force-based cooperation, localization before grasping and collision avoidance.

Considering the previous works presented in this section, most of the studies recognize that the UHF RFID technology is helpful in the navigation of mobile robots in indoor environments. In addition, we can see that the human–robot interaction is a valuable source of information for the robot to complete its navigation tasks. The present research improves the existing works in two ways, mainly: (i) We formally introduce the concept of waymarking in robotics. Although other studies explore communication between robots to aid navigation, there are no existing studies that allow the robot to rewrite or update the information on “marks”, in our case RFID tags, to signal the environment or give clear navigation instructions. (ii) We enrich the data of the RFID navigation system with the interaction through voice; in this work, the user is a crucial factor in obtaining valid information from the environment where the robot navigates.

This work extends our previous research [27,58], we presented a signaling and navigation system for indoor environments using UHF RFID technology; in that research, the signals’ structure and the searching process were designed and implemented. In this paper, we propose a waymarking system for updating those signals using the information provided by a real user.

## 3. System Overview

In this work we have used the Maggie robot [26]. Maggie is a social robot designed to have an attractive appearance for the users. Maggie’s movement is possible thanks to a differential wheeled base. For navigation tasks uses a Sick LMS 200 laser sensor. Moreover, Maggie has two UHF RFID readers that have been placed on both sides of its body (Figure 1).

### 3.1. Software Architecture of the Robot

In the literature, we can find software architectures applied to social robotics [59], some of them applying mental and biologically human-inspired models [60,61]. In [62] by example, researches use cognitive models based on [63] to connect language, perception, and action-related modules.

The software of the Maggie robot is based on the automatic-deliberative architecture, developed in the RoboticsLab of the UC3M [64,65,66,67]. Human mental processes inspire this approach. Figure 2 shows the diagram of the AD architecture and its different levels:

The automatic level refers to the automated or reactive processes of the system. This level gets and sends data to the robot’s sensors and actuators.Deliberative level is associated with reflective processes, namely, processes requiring reasoning or decision capacities.Communication between the automatic level and the deliberative level is bidirectional, through the short-term memory (STM).The short-term memory—also referred to as working memory—stores the most important data from the sensors and shared data among skills.A skill is the capability of the robot to perform reasoning, to process information or to carry out an action. Automatic and Deliberative levels are composed of skills.Skills can be triggered by other skills or by a command sequencer, returning data or events to the element that triggered them or to skills interested in the information provided.Communication between skills is done through events and the use of short-term memory. A skill can write data on STM and notify other skills—using events—availability of such data.The long-term memory stores the permanent knowledge. The robot uses this knowledge to reason or to make decisions.

Based on the previous specifications, the signaling and waymarking system presented in this work, will be developed on the basis of three conditions:It must be an AD architecture skill.It must share and receive information from other skills of the system (use of STM).Warn and receive events and notifications from other skills.

### 3.2. RFID Signage System

The RFID signage system model, the specification of the environment knowledge and the navigation algorithm have been presented in [27,68]. The basic element of the signaling system is the signal. A signal is defined as a type of sign situated in the environment to identify an element or a place, it also can be used to give an advise or indication. A signal should help a robot answering questions such as: Where am I?, How can I get there? and, Where can I go?.

The signal representation is based in the usually found in public buildings [69,70]. In Spain, most of the public buildings use a classification based on zones. In order to facilitate the identification and the relationship among places, as a general rule, each zone has an alphanumeric identification (Figure 3).

For example, when we refer to the *Office 1.3.B.15* at UC3M, the place will be formed by the following zones:$z_{1}$ = Office *1.3.B.15*;$z_{2}$ = Zone B *1.3.B*;$z_{3}$ = Third floor *1.3*;$z_{4}$ = Betancourt Building *1*;

The entities or elements that define a signal are:Zone: Refers to a sub-region or part of an environment area. One or more nested zones form a place. Each zone has a name and an alphanumeric identifier.Place: Space, place or entity that may consist of one or more nested zones.Connection: a connection is defined as the possible place or places that the robot can get to from its current place. A connection is formed by: a place, or a list of places, and an action.Action: It indicates the skill, or set of skills, that the robot must execute to get the place indicated by the connection. The action can be:
Topological: such as “turn right”, “move forward”, or “keep to the left”.Geometrical: the indication may tell the robot to go to a specific coordinate point; in this case, the signal may be an url address with a geometric map.

In summary, a *signal* $s_{0}$ is formed by a *place*
$p_{0}$ and a *connection*
$c_{i}$ or a list of connections. A connection $c_{i}$ indicates the place $p_{i}$ which the robot can get to from place $p_{0}$ indicated by the signal by executing an action $a_{i}$ (Figure 4).

Therefore, a RFID tag store the signal information: place, connections, and actions. The basic structure of a signal is modeled using an XML (eXtensible Markup Language) file. Figure 5 shows the definition of a place and a connection in XML. This data structure will be stored on RFID tags.

For the robot navigation, in this research, signals will be previously placed in the environment, at points that are considered key for the navigation and identification of places, such as intersections, doors, main entrance, hall, zone changes, etc. The tags may initially be empty or have information about the signal. When the robot finds a signal (in this case, an RFID tag), it will be able to add and/or update the information contained in the signal. The signals will be changed by the robot using the pattern o model of a signal.

To start the signaling process, the robot must acquire the information to write to the signal. The sources of information can be very varied, from asking a person to reading it on another signal and deducing what to write. This work focuses mainly on the acquisition of information through human–robot interaction and the writing/updating of data stored in the signals.

## 4. Skills RFID

The skills are the essential elements in Maggie’s software architecture. A skill is an ability to perform robot reasoning, process information or carry out an action. In software engineering terms, a skill is an object that encapsulates the data and processes that describe the robot behavior.

To develop RFID identification skills, two UHF RFID readers with built-in antenna have been incorporated into the robot. The robot will be able to read and write information on RFID tags that are nearby in the in the environment.

Two general RFID identification skills have been developed for reading and writing data. The data writing skill will be key in the waymarking process.

The RFID reader runs constantly detecting tags in its immediate area. RFID skills will notify other skills in the system about the detection, reading and writing of tags. It is also possible that other skills to send data to the RFID skills to be written to the tags.

Figure 6 shows the flowchart of the RFID read skill. When a sequencer or another skill triggers the skill, it starts searching for tags in the environment. If the tag is detected, it performs the reads the data stored in the tag and stores it in short-term memory. It immediately sends the event *NEW_TAG_RFID*, the skills interested or subscribed to it, can perform the corresponding actions. If the skill is still active, go back to search for tags and do the previous process again. the previous process, if it is not active, it finishes its execution or it is blocked waiting for orders.

The RFID writing skill is simpler. When it receives the write command, the skill activates the function *search tag*. If it finds the tag, it writes the data and finishes the write function. If it does not find the tag, it continues searching for it until it finds it, until a maximum search time elapses, or until it receives a command to interrupt the operation (Figure 7).

## 5. Information Acquisition Using HRI

In humans, it is common to ask other people how to get to a specific place if we don’t have a map or can’t find a sign to show us how to get there. Robots must deal with navigating in structured, indoor environments where people are part of the environment, such as homes, hospitals, museums, offices, etc, It is important to be able to combine the robot’s navigation tasks with human interaction. Using HRI, the robot can acquire valuable information from the environment to successfully reach its goal.

One of the objectives of this work is for the robot to be able to update the information of the signaling system using data that people in the environment can provide. In order to exchange enriching information, the robot will use verbal communication.

### 5.1. Human Robot Interaction Cases

The basic information in a signal must answer at least one of the following questions: (i) Where am I? (ii) How am I going? and (iii) Where can I go?.

If, for example, the robot is navigating in the environment and finds an empty sign, to acquire that information, the robot asks and receives the indications of a person to signalise the environment. To query and respond, the robot must use natural language and not a technical language, that is, the robot cannot ask questions like “Tell me the content of the label <place>”.

Dialogues between human and robot must be understandable by both parties, on the one hand that the person can respond with a natural language and can understand the robot consulting, and on the other hand, the robot must be able to understand the information received, and then process it and write the RFID signal.

In this paper, we handle two possible cases for starting the interaction: (i) the robot finds a signal and asks a user for the missing information and (ii) the user shows the robot the environment to write the RFID signals. This combination of two processes is one of the advantages of the presented solution compared to other research in this area. The versatility of the RFID signals that can be adapted and extended with information acquired by the robot’s sensors and by the information given by the users. Each of the cases is detailed below:

#### 5.1.1. Case 1: Robot Takes Initiative

A simple case of data acquisition would consist of a simple dialogue, which would start when the robot finds an empty signal. If the robot is close to a person, the robot can take the initiative and request, through voice, the data concerning the place where it is located and write it on the signal.

It is also possible that the robot, during the conversation, ask for information about the place where it is located and the connections or places where it can go from that place with the corresponding actions. The robot will ask until the user tells it that he/she has no more information, and it is even possible that the user provides incomplete information, for example, that it knows the name of the place but not its identifier, or that the user does not know the name of the current place but knows where the robot can go. Figure 8 shows the sequence diagram for case 1.

#### 5.1.2. Case 2: User Takes the Initiative

Another possible case is when a person takes the initiative and indicates to the robot the content of the signals. The person can show the robot where the signals are located, once the robot is at the signal (in this case an RFID tag), the user proceeds to give the robot the information of the place, through natural language, which the robot will process and model to write it in the signal. The user can guide the robot to the signals through movement commands. The robot will execute the indicated skill, through phrases such as “Follow me” or “Turn right”, until it finds a signal. Figure 9 shows the sequence diagram for case 2.

## 6. Grammars for Signaling Interaction Dialogues

The design of the dialogues is one of the bases of this work, through these dialogues, the robot will be able to communicate with the user and to acquire the information it needs to modify the information of one or several RFID signals of the environment. To achieve this purpose, we have used two skills developed and integrated into the software architecture of the robot: An auto-automatic speech recognition skill and a text-to-speech synthesis skill [71,72,73]. For dynamic dialogues, the system uses a dialogue manager. The dialogue manager is based on a VoiceXML interpreter and its objetive is to connect the speech skills to each other to keep the coherence during the interaction [72].

In grammars, the elements that constitute the language are established, so that the speech recogniser can relate words to semantic content [74]. Grammars, in this work, refer to phrases and/or words that the robot will recognise and associate it with the environment information and the navigation tasks. The grammar will depend on the name of the place where the robot can be found, the places where it can go and the actions it must carry out.

### 6.1. Possible Sentences

Grammar design is based on the possible sentences that the user can give to the robot to indicate the information of a signal. The sentences should answer the basic questions: Where am I? Where can I go? and How can I go? For signaling dialogues, there are different types of possible sentences:Description of a specific place: Refers to the specific places name. It will depend on where the robot is navigating. For example: *‘Director’s Office’*, *‘Robotics Lab’*, *‘Computer Classroom’*, etc.Description of current location: Include instructions such as: *“You are in…”*, *“This is…”*.Unique information about a place: Refers to the unique identifiers of each place. For example: For the *‘Robotics Laboratory 2’* the ID will be *‘1 point 3 C 13’*.Directional sentences or connection-related sentences: Include instructions such as: *“You can go to…”*, *“It’s this way”*, *“over there”*, etc.Description of a specific action: Related to the robot’s navigation skills to get from one place to another. For example: *“turn right”*, *“go straight”*.

### 6.2. Language Definition

Based on the user’s possible sentences, the language is defined with the different grammatical rules, the robot will relate these rules with the signaling task:$places: Places or zones to be signaled and/or to where the robot can go.$ids: Alphanumeric identifiers for places or zones in the environment.$actions: Movements or tasks to be executed by the robot in order to reach an area or place.$adjectives: Directional adjectives that complement the location and connection information.$type_of_action: This rule will determine if the action is geometric—it requires a geometric map—or not.$op: Generic answers of affirmation or negation.

### 6.3. Semantic Definition

The semantic grammar allows the recognition system to return values related to the meaning of the sentences it detects. According to the ABNF standard (augmented Backus–Naur form) [75], it is possible to incorporate a semantic interpretation within each grammar rule:
<@attribute=value> where @attribute is the variable that stores the semantic value given by value.

    For example, in the following grammar rule:


*public $op= (“yes”:yes|“no”:no|“ok”:yes|“negative”:no {@option $value}*


The variable *@option* contain the semantic value of the grammatical rule *$op* (*“yes”, “no”, “negative”…*). This variable will be used by the dialogue manager to store the recognised phrase and send it later to the robot’s signaling skill.

### 6.4. Placewaymarking, Optionwaymarking and Connectionwaymarking Dialogues

The set of dialogues works like a state machine, where each dialogue specifies the information given by the robot to the user and vice versa. Depending on the answers received or sent, the system is passed on to the next state.

The activity diagrams for each dialogue are explained below, these diagrams show the different phases of the dialogue to get the data of a signal.

#### 6.4.1. Placewaymarking Dialog

In Figure 10, the behavioral scheme of the first dialogue *dialogPlaceWaymarking* is presented. The robot gets information about the current location (i.e., where the signal is located). At the start, the events that will notify the system of the availability of new data in short-term memory are defined. Then grammar related to waymarking skills is assigned. Once this is done, the form is executed. The robot starts requesting information about the characteristics of the place where the sign is located: the name of the place and its ID. The robot stores the obtained data in the Short-Term Memory and notifies the system by emitting the event (*EVENT_PLACE*). Subsequently, the robot asks the user if she/he has more information from that place, if so, it starts the dialogue again, this is intending to complete the information in the case that the *place* is composed of nested areas. In case the user has no further information the system proceeds to the next dialogue (*dialogOptionWaymarking*).

#### 6.4.2. Optionwaymarking Dialogue

When the place data has been acquired, the robot proceeds to ask about the existence of connections. If the user does not confirm the possibility of going to other places, the system ends the dialogue and calls the *dialogSaveWaymarking*, otherwise, if the user confirms the existence of connections, the system proceeds to the next dialogue *dialogConnectionWaymarking*.

Figure 11 shows the activity diagram of the *dialogOptionWaymarking* dialogue.

#### 6.4.3. Connectionwaymarking Dialogue

If the user has information about other places that are possible to go from the location of the signal, the *dialogConnectionWaymarking* dialogue is executed. Figure 12 shows the activity diagram of this dialogue.

Initially, the grammar related to waymarking skills is assigned. The robot then proceeds to query the data for each connection. It asks for the name and the ID of the connection. The questions are simple, similar to the ones shown in the dialogue of information acquisition from a location.

As the connections represent places, they may contain nested information (nested zones), so the robot will ask until the user confirms that it has no more information for the particular connection. When the robot has acquired the information, the dialogue manager will store the information in the short-term memory of the system and emit the event *EVENT_CONNECTION*.

The next part of the dialogue complements the connection information. The robot asks the user for the action to be performed to go to the indicated place. This type of action may be topological—“*turn to the left*”, or geometric—“*go to (x, y) point*”, if the action is geometric a geometric map may be required.

Once the action information has been completed, the dialogue manager stores the information in the short-term memory and an event to notify the skills or processes interested in the data.

The system repeats the process if the user reports the existence of other connections. If the user has no further information, the dialogue manager notifies the system using the *EVENT_CONNECTION* event.

## 7. Signaling Skill

The objective of the signaling Skill (*CSkill_Waymarking*) is to manage the information acquired through human–robot interaction and to format it into an appropriate human–robot interaction and format it in a suitable format to write it into a signal.

The signaling skill stays listening to the events emitted by the Dialogue Manager and It will read from the system’s short term memory, the place and connection data provided by the user.

Figure 13 shows the schematic of the signaling system with the signaling skill.

The signaling skill mediates between the Dialogue Manager and the writing skill *CRFID_WriteSkill*. It subscribes to the events emitted by the dialogue manager. These events notify that a signal data has been stored in the short-term memory. The events are:DATA_VXML_PLACE: Notifies that information from a place has been stored. This event is sent whenever the dialogue manager stores information for a specific zone of a place. If the place $l_{i}$ is formed by *n* zones, so that: $l_{i}=\left\{z_{1},\cdots z_{n}\right\}$, then this event is emitted upon receiving information from each zone $z_{i}$.Upon receiving the event, the skill reads from short-term memory the following information:
–Label: nName of the zone.–ID: alphanumeric identifier of the zone.–Option: variable to determine if the information of the *n* zones of the corresponding place $l_{i}$ has been obtained.DATA_VXML_CONNECTION_PLACE: Notifies that connection data has been stored. A connection is formed by a place $l_{c}$ and an action $a_{c}$. This event then notifies the information of the place $l_{c}$ that indicates the connection. The data received corresponds to the schema of a place:
–Label: corresponds to the name of the zone that indicates the connection.–ID: alphanumeric identifier of the zone.–Option: variable to determine if the information of the *n* zones has been obtained from the corresponding place $l_{i}$ that indicates the connection.DATA_VXML_ACTION: Notifies that the action information has been stored.The signaling skill on receiving the event reads from short-term memory the following information:
–Label: name or description of the action.–Type: type of action: topological or geometric.EVENT_ENDING_CONNECTION and EVENT_NO_ENDING_CONNECTION: These events allow the skill to know if all the place connections have been stored in the short-term memory. In other words, the system needs to know when it has finished sending the connection information. The dialogue manager notifies it when the user tells it that it does not know or cannot tell it other places where it can go from the place where the robot is.

### Signaling Skill Behavior

Depending on the event received, the signaling skill *CSkill_Waymarking* will have different behavior:If it receives the DATA_VXML_PLACE event, the skill will read data from a place in short-term memory.If it receives the event DATA_VXML_CONNECTION_PLACE, the skill will read partial data from a connection (where it is possible to go).If it receives the event DATA_VXML_ACTION, the skill shall read, from short-term memory, the *action* that must be performed to go to the place indicated in the connection.

When the skill *CSkill_Waymarking* gets the data, it gives it the format of a signal: place and connections.

For a place, the class store the data in a list of the nested zones.

For a connection, its two components will be stored separately: the place and the action or set of actions.

If the action is topological: It receives the name of the action, the type, an identification code and a control indicator to manage the list of actions.If the action is geometric: In addition to the parameters received in the case of a topological action, receives the address of an XML file with the geometrical map data and the geometrical location of the signal in the indicated map.

The signaling Skill emits an event to notify the signal write skill (*CRFID_WriteSkill*) that the data is available for writing. Then, data is available to be physically written to the signal—RFID tag.

## 8. Experimental Results

This section presents the dialogues to test the human–robot interaction signaling system. The goal of each dialogue is that the robot can acquire the information necessary to signal the corresponding RFID tag. The examples below aim to signal the Third Floor of the Betancourt Building of the University Carlos III of Madrid (Figure 14).

### 8.1. Semantic Grammar

The grammar allows the dialogue system to return values related to the meaning of the phrases it recognizes. For this experiment, a grammar describes the information about the places and connections to signal the Third Floor of the UC3M Betancourt Building. The grammar was initially designed in Spanish and, for this article, has been translated into English.

Below shows an abstract of the rules created for the places and connections of the environment is shown. The semantic grammar has been designed based on the grammar rules explained in Section 6.2.

$root = $waymarking;$waymarking = $adjectives | $ids | $op | $actions | $codes | $typeofaction;$places = ("Betancourt Building" | "third floor" |  "Robotics lab" |                "TV" | "zone" | "toilets" | "office"){<@places $value>};$ids = ("one"  | "1 point 3"  | "1 point 3 c" | "1 point 3 c 12"      | "1 point 3 c 12 tv"  ){<@ids $value>};$actions= ("follow me"  | "move to the left"  | "move to the right"           | "go to" | "go to point"){<@acciones $value>};$op = ("yes": yes | "no": no | "ok": yes |  "negative": no     | "alright": yes ){<@op $value>};

In $places, the possible names of places and zones to be marked are indicated. These are the places where the robot can go in the environment where the experiment has been conducted.$ids are the alphanumeric identifiers of the places to be signaled. The identifiers must correspond to the building signaling standards of the Universidad Carlos III de Madrid.$op for yes or no answers.$actions are the phrases the user will say to the robot to indicate the movements or actions that it must execute to go from one place to another.

The interaction cases and dialogues have been designed to answer the following questions, which are necessary to configure the information of a signal: Where am I? Where can I go? and What do I have to do to get to that place?

### 8.2. Case 1: The Robot Takes the Initiative

This experiment is the simplest; the robot can take the initiative when it finds an empty RFID tag and asks the user for the information of that place and its connections. The robot is initially located in the corridor on the third floor, next to the office 1.3.C.11 (Figure 15 and Figure 16).

#### 8.2.1. Where Am I?: DialogPlaceWaymarking

In the first example, dialogue is related to requesting information about the current place. The robot makes a simple query about its location, asking for the elements of a place: the name and the identifier.

The dialogue shown specifies an informal way of asking for information about where the robot is. The user’s answer will be related to possible places in the environment previously specified in the grammar. In this example, the user replies to the robot with no more information, and the dialogue ends. Figure 17 shows the dialogue and the XML data generated to write in the signal (tag RFID).

The dialogue can be enriched when the robot request for information about nested zones in the place. Then the user will provide the robot with additional information about the place where it is positioned. Figure 18 shows the dialogue when the robot asks for the nested zones of the current place and the XML data generated.

#### 8.2.2. Request for Additional Information: DialogOptionWaymarking

In this dialogue, the robot asks the user if he/she has additional information for signaling. It is designed to serve as a connection between the “Where am I?” dialogue and the “Where can I go? dialogue.

Depending on the user’s answer, two cases can occur, if the answer is positive, the next dialogue will be activated (to get information on the connections), and if it is negative, the dialogue ends.

Figure 19 shows an example of the possible dialogues, depending on if the user’s answer is positive or negative.

#### 8.2.3. Where Can I Go?: DialogConnectionWaymarking

This dialogue allows the robot to get the connections or places where it can go, from the site where it has found the signal. The user will be able to give the number of connections known to the user.

The robot can ask user for information about a connection. The user can indicate to the robot the nested areas of the place where it can go from the site that is being signaled. Below is an example of this type of dialogue.

Figure 20 shows the dialogue and the XML data generated to write in the signal (tag RFID).

Once the place information of a connection has been completed, the robot queries information about the action or set of actions it must do to go to the place previously indicated.

Additionally, in the dialogue, once the information of a connection is completed, the robot can continue consulting the user for other connections, that is, for other places where it can go from the signal that is being written. So the robot will ask the user for the existence of these. If the answer is affirmative, the same pattern of the dialogues seen above is repeated. An example of such a dialogue is shown in Figure 21.

### 8.3. Case 2: The User Takes the Initiative

In the previous dialogues, the robot takes the initiative in the human–robot interaction when it is next to a signal (RFID tag). The user takes the initiative and provides the robot with information about the signal to be written. The user then guides the robot through the environment to reach other RFID tags.

#### 8.3.1. The User Indicates to the Robot the Current Place

In the following dialogue, the user takes the initiative and tells the robot where it is (name and identifier). The place does not have nested zones.

User: Maggie, follow me.Robot: Perfect.....Robot: I have detected a tag.User:  All right Maggie;  I’m going to give you directions to this place.....Robot: Then I’ll save the information.User: Now follow me, I’ll keep showing you around. Robot: Good.User: Maggie, I’m going to give you the directions to this place. This place is...

Similarly, as seen in the Section 8.2.1, the user will be able to give information to the robot about the nested zones, with the difference that the robot does not ask for the info but waits for the user to indicate it.

#### 8.3.2. The User Indicates Movement Actions to the Robot

The importance of the robot being able to signal the environment (write information on the RFID tags) is directly related to the automation of the system. The user will interact with the robot with colloquial dialogue and “show the robot the environment” to store the information on tags distributed in the environment.

The dialogue used for this purpose will look like the one shown below.

User: Maggie, follow me.Robot: Perfect.....Robot: I have detected a tag.User: All right Maggie.  I’m going to give you directions to this place.....Robot: Then I’ll save the information.User: Now follow me, I’ll keep showing you around.Robot: Excellent.User: Maggie, I’m going to give you the directions to this place. This place is...

As mentioned above, the robot must be attentive to the information of the signals and the movement indications. The robot waits for the user’s indications. Once it receives the movement action commanded by the user, it sends this information to the system’s short-term memory so that an action manager can send the movement command to the corresponding skill.

In Figure 22, an example of signaling is shown. The user gives movement commands to the robot to signal commands to the robot to signal some tags on the third floor of the UC3M Betancourt Building.

The user starts the route with the robot at signal 1. The user tells the robot the information of the place *Office-1.3.C.11*. Subsequently, the user gives the robot the movement command *“Follow me”* to reach signal 2.At signal 2, the user indicates to the robot the information of the place *“Maggie’s lab-1.3.C.12”*. The robot stores the data and waits for a new command from the user. The user commands the robot to *“follow the wall to the right”*.At signal 3, the user tells the robot the information of the place *“Robotics Lab 2-1.3.C.13”*. The robot stores the information and waits for a new command from the user. The user commands the robot *“follow the wall to the right”*.Finally, when the robot detects signal 4, it waits for the user to give it the information of the place *“Robotics Lab 3 -1.3.C.14”*. When the user finishes giving data to the robot, the robot stores the information and waits for the user’s command to finish executing the signaling skill.

## 9. Conclusions and Future Works

Usually, when talking about navigation in robotics, the robot’s interaction with the environment is limited just to get information, the robot only reads or perceives objects and signals, but it is not an agent of change of the environment. Generally, changing the information of a navigation signal is a task usually left to humans.

In this work, we present an automated environment signaling system using human–robot interaction and RFID technology. We introduce the term waymarking as the skill of a robot to signal the environment or generate information to facilitate localization and navigation. In this process, human–robot interaction plays a key role. A robot can interact with people to obtain the signals information and thus automate the waymarking process.

Our main objective focused on two issues:The robot should be able to update the signals (RFID tags) of the environment signaling system,The robot should be to interact with people to request/receive the information to write/update each signal.

The contributions of this research include a system to write signals in indoor environments. Skills were designed and programmed in the robot such as writing RFID skill and the signaling skill. In addition to physically writing information to the RFID tags, the developed algorithm coordinates the movement commands a person gives to the robot.

The robot must acquire the information that it will later write on the signal to carry out the signaling process. For this purpose, human–robot interaction was added to the system. Through verbal communication, the robot communicated with users. A grammar was designed based on the possible sentences that could be received by the user and translated into the context of each RFID signal. Two situations were identified, depending on who takes the initiative: the robot or the user. The dialogue system was successfully integrated into the robot’s software architecture, sharing information through the system’s Short Term Memory and emitting/reacting to the events of the designed system.

Finally, this interaction signaling system was successfully implemented in a real social robot in a structured indoor environment (Betancourt Building—Carlos III University, Madrid, Spain). The tests carried out to prove that the naturalness with which the robot requests or receives the information makes the process of writing the signals more comfortable without the need for the user to know about RFID technology. The robot uses common phrases such as “Where am I?” and “Where can I go?”, just as we humans do when we ask other people for information about the environment. It is also possible to guide the robot and “show” it the environment to carry out the task of writing the signs.

As future work, we consider adding the learning factor by the robot; using artificial intelligence tools, the robot could go through the environment and learn about possible changes based on the signals previously read or previous interactions with users. In this line, it would be favourable to incorporate semantic navigation concepts, where each element has semantic meaning for the robot.

Another possible improvement will be to test the signaling system in other environments to check its performance and robustness, such as in hospitals, shopping centres, museums, etc. One of the barriers is the increment of the dialogues complexity and the spatial description, studies as [76,77] are considered a good option to improve the system.

We also consider it interesting to add communication through gestures- e.g., recognise arm movements indicating *“go this way”*, analyse the effect of embodiment and make the interaction more inclusive.

To do this, it will be necessary to re-evaluate and upgrade the sensors used in this work, e.g., a newer laser sensor and the incorporation of cameras for advanced vision.

Last but not least, to continue this research properly, a quantitative and qualitative evaluation of the results should be carried out; this will help us get more out of the experiments for further studies.

## Figures and Tables

**Figure 1 sensors-21-08145-f001:**
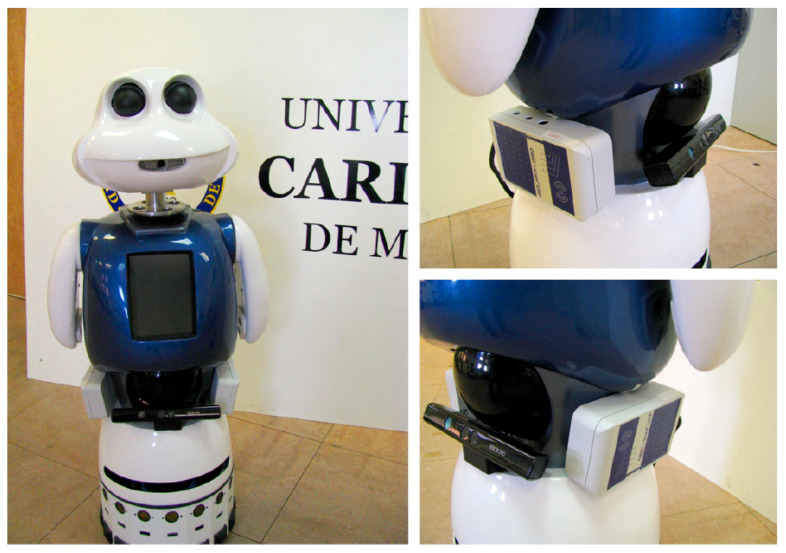
The social robot Maggie and its two RFID UHF readers/writers.

**Figure 2 sensors-21-08145-f002:**
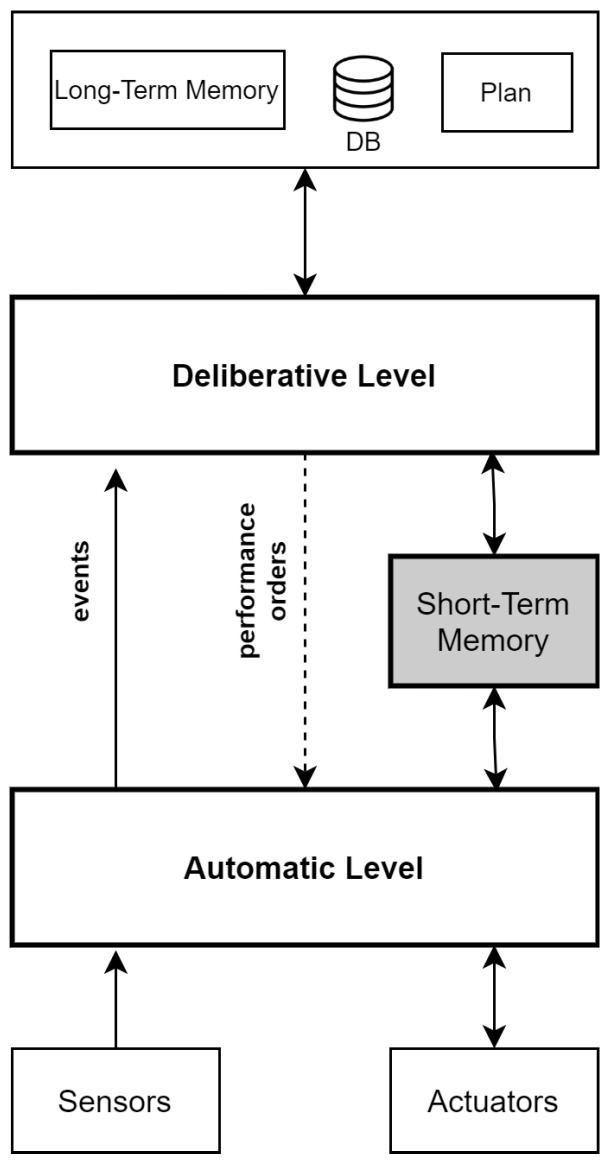
Automatic-deliberative architecture.

**Figure 3 sensors-21-08145-f003:**
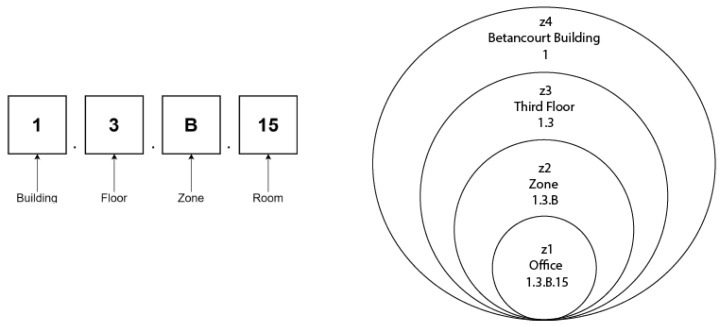
Example of an alphanumeric code for office 1.3.B.15 at Universidad Carlos III de Madrid and its representation with nested zones.

**Figure 4 sensors-21-08145-f004:**
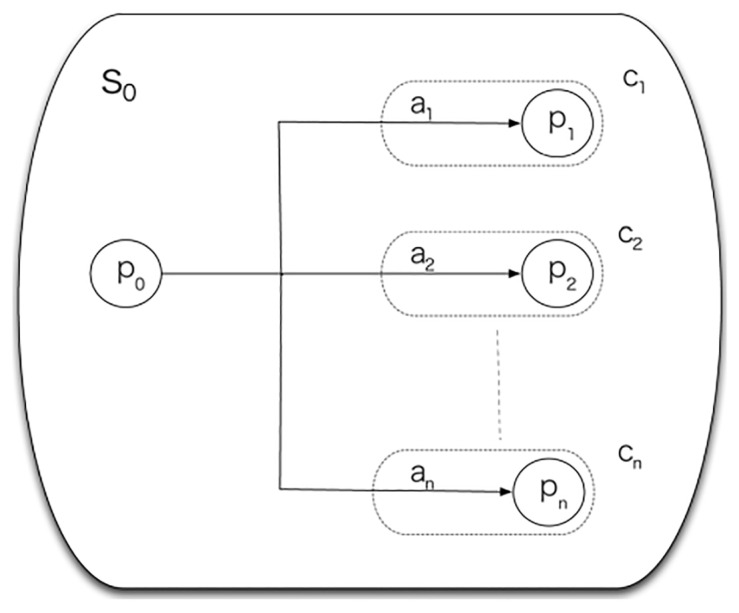
Representation of a signal.

**Figure 5 sensors-21-08145-f005:**
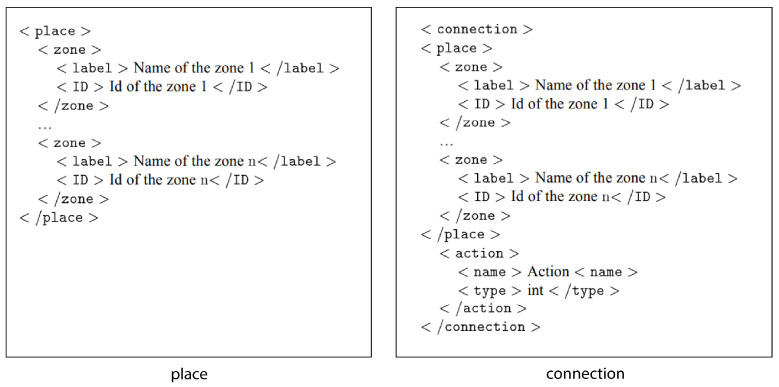
XML representation of a place and a connection.

**Figure 6 sensors-21-08145-f006:**
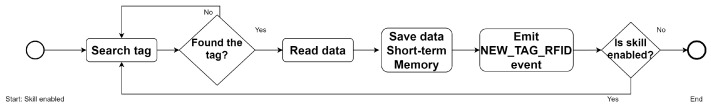
RFID reading skill flowchart.

**Figure 7 sensors-21-08145-f007:**
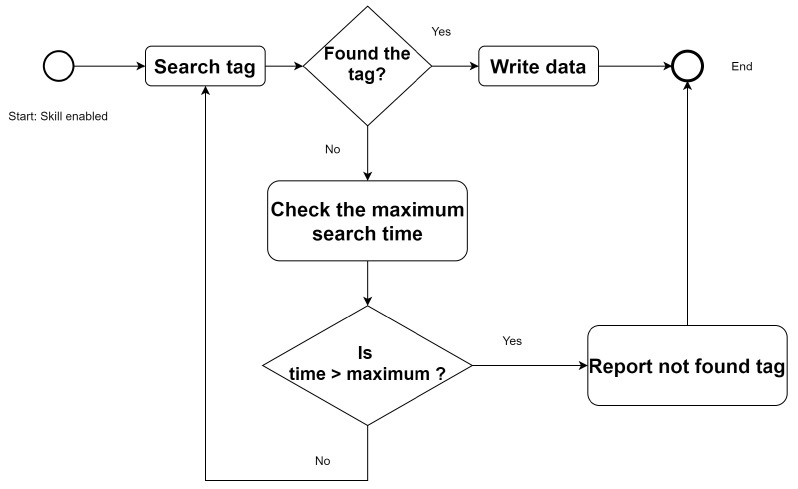
RFID writing skill flowchart.

**Figure 8 sensors-21-08145-f008:**
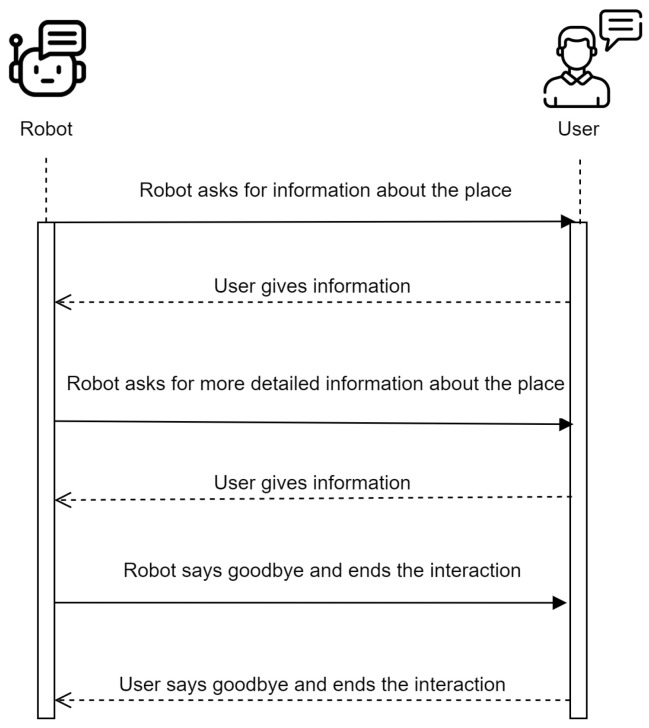
Sequence diagram: Where am I?

**Figure 9 sensors-21-08145-f009:**
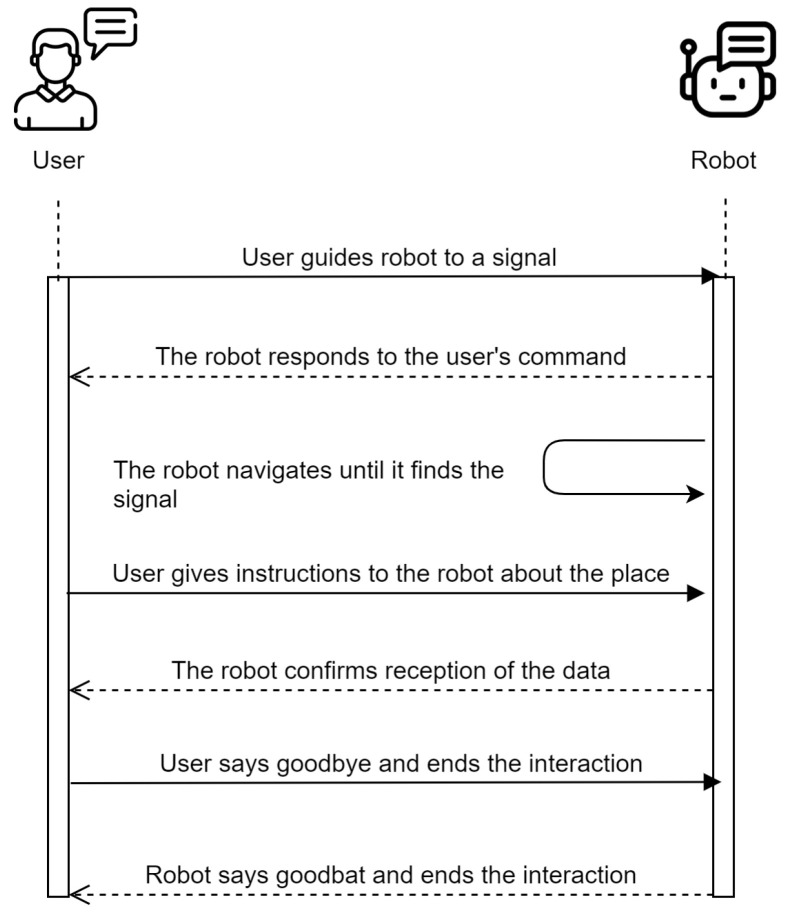
Sequence diagram: user takes the initiative.

**Figure 10 sensors-21-08145-f010:**
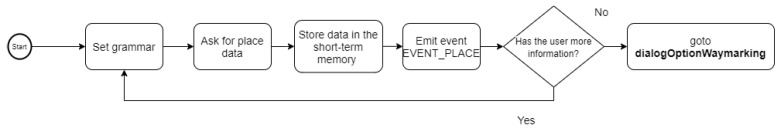
Activity diagram: *dialogPlaceWaymarking*.

**Figure 11 sensors-21-08145-f011:**
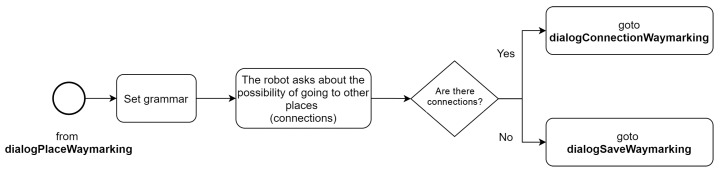
Activity diagram: *dialogOptionWaymarking*.

**Figure 12 sensors-21-08145-f012:**
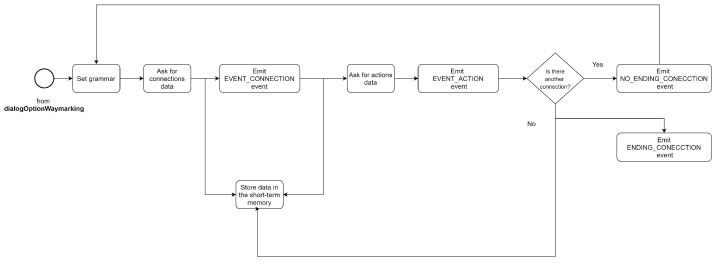
Activity diagram: *dialogConnectionWaymarking*.

**Figure 13 sensors-21-08145-f013:**
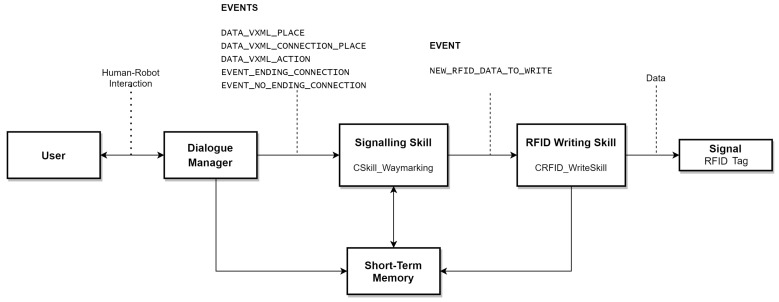
Diagram of the signaling system and its different elements.

**Figure 14 sensors-21-08145-f014:**
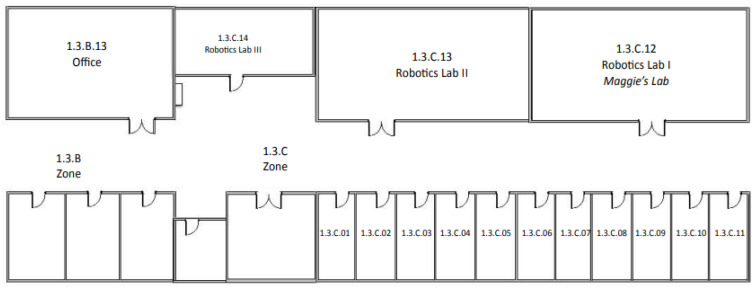
Third floor of the Betancourt Building of the University Carlos III of Madrid.

**Figure 15 sensors-21-08145-f015:**
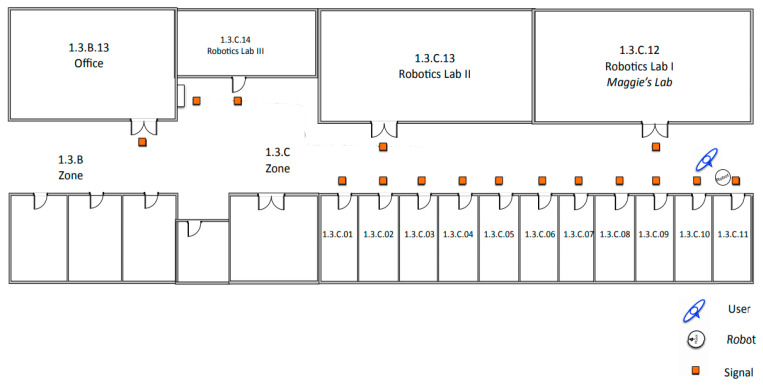
Robot and user are next to the the office 1.3.C.11.

**Figure 16 sensors-21-08145-f016:**
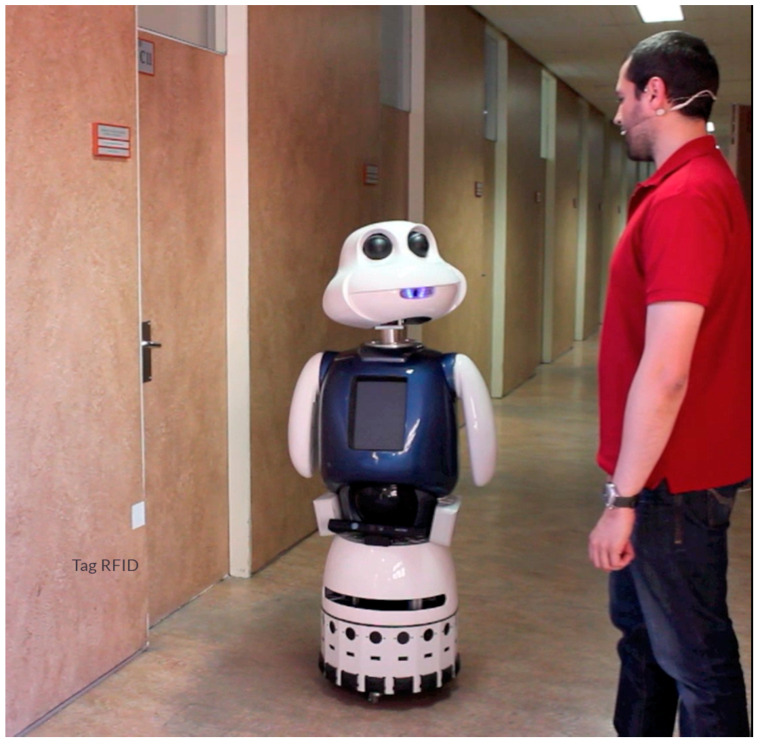
Robot–human interaction during waymarking process.

**Figure 17 sensors-21-08145-f017:**
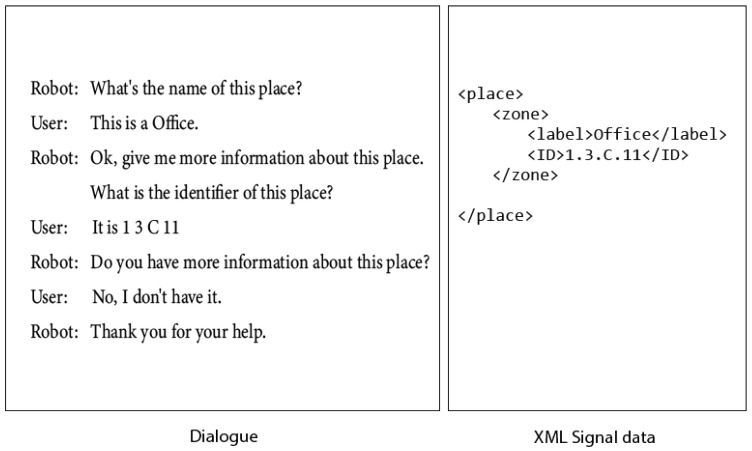
Case 1: The robot takes the initiative in the interaction. Dialogue and XML data generated.

**Figure 18 sensors-21-08145-f018:**
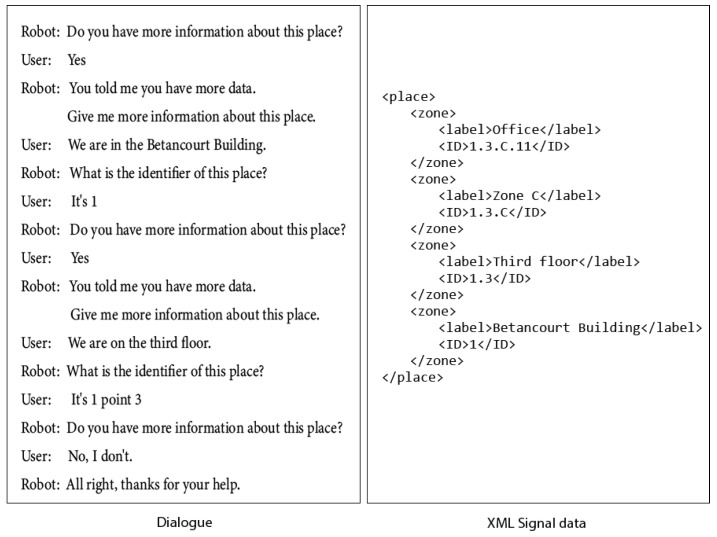
The robot request for information about nested zones. Dialogue and XML data generated.

**Figure 19 sensors-21-08145-f019:**
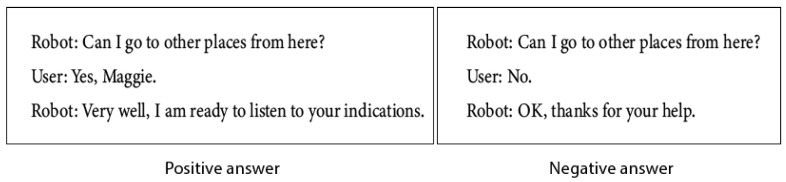
Request for additional information.

**Figure 20 sensors-21-08145-f020:**
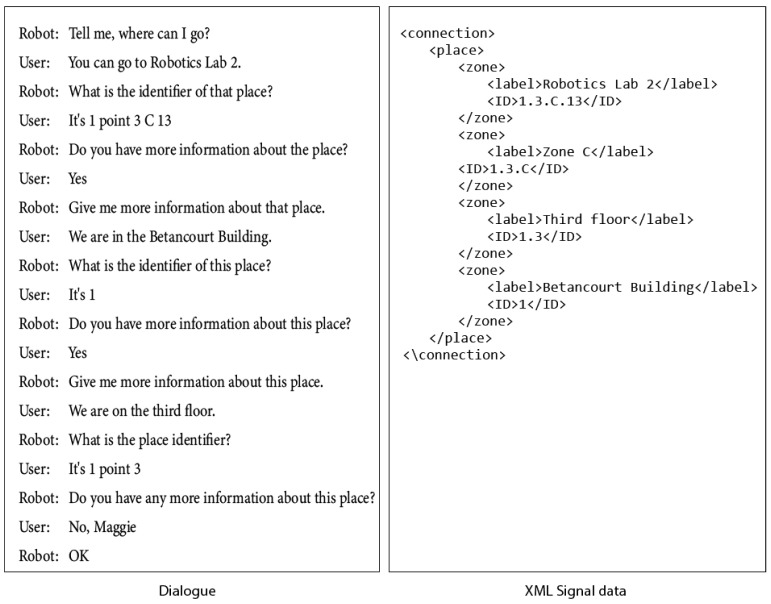
The robot ask user for information about a connection. Dialogue and XML data generated.

**Figure 21 sensors-21-08145-f021:**
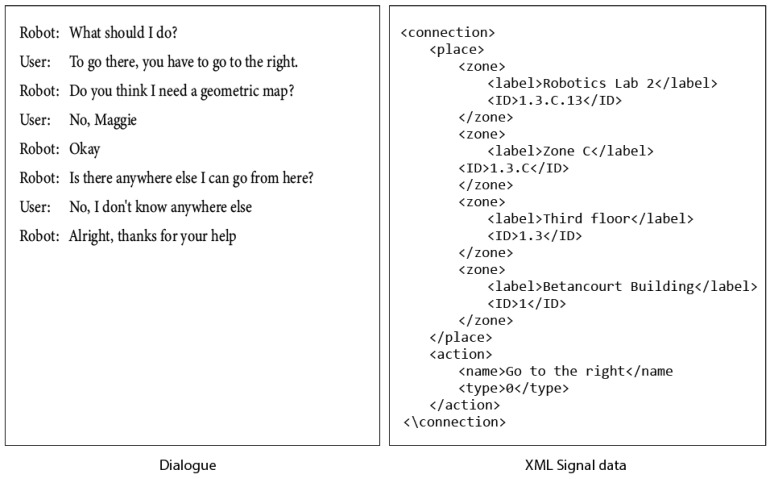
The robot ask user for information about the action. Dialogue and XML data generated.

**Figure 22 sensors-21-08145-f022:**
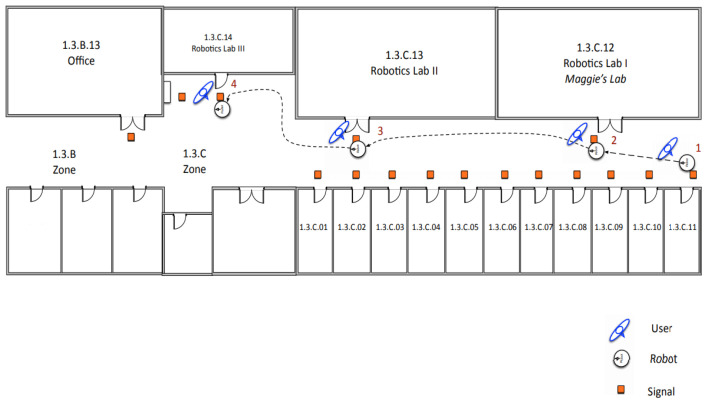
Example of environment signaling. The user gives movement commands to the robot.

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
