# Peer review of "Waymarking in Social Robots: Environment Signaling Using Human–Robot Interaction"

_sensors, 2021, doi:10.3390/s21238145_

Round 1

Reviewer 1 Report

The article is interesting and the problem is worth exploring, but it is recommended to complement the discussion and conclusion section with an indication of what is innovative in the presented solution in comparison to other such solutions (authors' contribution to the development of science).

It would also be good to characterise the barriers that the authors encountered in their experiments with the robot and point out the advantages of the presented solution compared to other research in this area.

Author Response

Dear Reviewer 1, 

Reviewer 2 Report

The paper presents adequately original and directly applicable work on a system accomplishing robotic navigation by means of verbal Human-Robot Interaction as well as RFID tags – utilizing what they call “waymarking”. One of the most positive aspects of the paper is its very high clarity of exposition as well as clarity of expression. Its main shortcomings are two: First, its background references can be improved, also by slightly widening their context – so things can be viewed under somewhat wider, yet highly relevant, themes. Second, what is missing is a real qualitative/quantitative evaluation; however, I believe that this should be left as future work, as I think that what is presented in this paper is already clearly adequate for publication.

Regarding the wider context: First of all, from the moment that an LMS200 LRF exists on the robot (as is the case in many other robots; and nowadays there exist increasingly cheaper and better LRF’s), and also given the very low price and high availability of camera-based vision system and off-the-shelf advanced plug-and-play computer vision, one would wonder why these are not utilized in conjunction with what is presented in the paper. Of course, in terms of clarity of story and exposition, the presented system certainly achieves these two goals; and a more complex system would be less crystal-clear; still yet, at least at the level of discussion and future plans, further details on the integration of LRF/advanced vision to what is described, would be welcome.

Second, regarding the connection between low-level control loops, landmark locations, all the way to trajectories and maps, qualitative or quantitative/metric, a seminal piece of work that is worth citing (as well as studying and contextualizing to your work!) is Benjamin Kuiper’s “Spatial Semantic Hierarchy” [1][2]. Third, beyond the (quite poor, but adequate for the purposes of the paper) right/left minimal set of spatial descriptions, one should at least mention the intricacies of expansion to a wider set of spatial descriptions, including the complexities of the semantics of spatial prepositions, all the way from those visible in early corpora such as the HCRC Maptask [3] and the follow-up versions of it from other institutes (such as SMTC-A[4]) as well as its descendants (such as ILMT-s2s[5]), to comprehensive reviews of existing models, notably such as [6]. Then, given that robots such as the ones that this work is targeted towards, will share spaces with humans, it becomes important to briefly mention work dealing with human-aware navigation and proxemics (a very recent survey from 2021 worth citing is [7]). Also, being able to have greetings with names and personalized dialogues, as well as indexical pointing (for example the arm movements indicating “go this way!”, can really help regarding Human-Robot Interaction for navigation: and one can easily support that since more than two decades these capabilities exist – through citing [8]. Regarding Human-Robot Interaction towards navigation, it would be worth noting that the humans need not be physically co-present: An example of robots interacting with humans through multiple channels, some of which physical and some virtual (real-time chat, for example through social network messaging, or even off-line posting and reading), which is worth citing, is [9]. Such techniques could potentially allow robots to quickly gain much more knowledge through cooperative humans, without needing physical co-presence.

Last but not least, another point of wider contextualization of the paper, is centered around cognitive architectures for robots: in Figure (2) you do present your Automatic-Deliberative architecture, but it would be great to also think about how your architecture could have a non-siloed approach towards connecting words (and spatial relations and referring expressions) to sensory information connected to places and routes. The classic way of doing so, which is inspired by the Cognitive Science concept of “Situation Models” [10], i.e. models of the situation in which the robot is embedded at the moment, or is remembering, or is imagining (while doing planning or doing story understanding, for example). And the direct application to robots is described in detail in [11]-[12], which would be worth citing, as relevant background, and also might be interesting when thinking about future extensions to your work.

Thus, in conclusion, this is a very well written paper, with clear relevance and with adequate novelty, which is worthy of publication. Consider doing a proper evaluation in a next paper (what you have here is enough), but certainly do add the suggested background references, in order to better contextualize your work – and thus, with these minor revisions, your paper will soon be ready for publication!

[1] Kuipers, B., 2000. The spatial semantic hierarchy. Artificial intelligence119(1-2), pp.191-233.

[2] Beeson, P., Modayil, J. and Kuipers, B., 2010. Factoring the mapping problem: Mobile robot map-building in the hybrid spatial semantic hierarchy. The International Journal of Robotics Research29(4), pp.428-459.

[3] Anderson, A.H., Bader, M., Bard, E.G., Boyle, E., Doherty, G., Garrod, S., Isard, S., Kowtko, J., McAllister, J., Miller, J. and Sotillo, C., 1991. The HCRC map task corpus. Language and speech34(4), pp.351-366.

[4] Helgason, P., 2006. SMTC-A Swedish Map Task Corpus. In Proceedings of Fonetik (pp. 57-60).

[5] Hayakawa, A., Luz, S., Cerrato, L. and Campbell, N., 2016, May. The ILMT-s2s Corpus―A Multimodal Interlingual Map Task Corpus. In Proceedings of the Tenth International Conference on Language Resources and Evaluation (LREC'16) (pp. 605-612).

[6] Zwarts, J., 2017. Spatial semantics: Modeling the meaning of prepositions. Language and linguistics compass11(5), p.e12241.

[7] Möller, R., Furnari, A., Battiato, S., Härmä, A. and Farinella, G.M., 2021. A Survey on Human-aware Robot Navigation. arXiv preprint arXiv:2106.11650.

[8] S. Malasiotis et al., “A face and gesture recognition system based on an active stereo sensor”, IEEE International Conference on Image Processing, ICIP 2001 Conference, Vol. 3, Thessaloniki, October 2001, pp. 955-958,

[9] Mavridis, N., Datta, C., Emami, S., Tanoto, A., BenAbdelkader, C. and Rabie, T., 2009, March. FaceBots: robots utilizing and publishing social information in facebook. In 2009 4th ACM/IEEE International Conference on Human-Robot Interaction (HRI) (pp. 273-274). IEEE.

[10] Zwaan, R.A. and Radvansky, G.A., 1998. Situation models in language comprehension and memory. Psychological bulletin123(2), p.162.

[11] Mavridis, N. and Roy, D., 2005, July. Grounded situation models for robots: Bridging language, perception, and action. In AAAI-05 workshop on modular construction of human-like intelligence.

[12] Mavridis, N., 2007. Grounded situation models for situated conversational assistants (Doctoral dissertation, Massachusetts Institute of Technology).

Author Response

Dear Reviewer,

Thank you for all your suggestions. 
